# Advances in Therapeutic Contact Lenses for the Management of Different Ocular Conditions

**DOI:** 10.3390/jpm13111571

**Published:** 2023-11-03

**Authors:** Mariana Ioniță, George Mihail Vlăsceanu, Alin Georgian Toader, Marius Manole

**Affiliations:** 1Faculty of Medical Engineering, University Politehnica of Bucharest, 011061 Bucharest, Romania； georgian.toader3007@upb.ro; 2Advanced Polymer Materials Group, University Politehnica of Bucharest, 011061 Bucharest, Romania; 3ebio-Hub Research Centre, University Politehnica of Bucharest-Campus, 061344 Bucharest, Romania; 4Department of Prosthetics and Dental Materials, Faculty of Dentistry, University of Medicine and Pharmacy “Iuliu Hatieganu”, 400012 Cluj-Napoca, Romania; marius.manole@umfcluj.ro

**Keywords:** contact lens, drug delivery, bioavailability, composite, gas-permeable, hydrogel

## Abstract

In the advent of an increasingly aging population and due to the popularity of electronic devices, ocular conditions have become more prevalent. In the world of medicine, accomplishing eye medication administration has always been a difficult task. Despite the fact that there are many commercial eye drops, most of them have important limitations, due to quick clearance mechanisms and ocular barrers. One solution with tremendous potential is the contact lens used as a medication delivery vehicle to bypass this constraint. Therapeutic contact lenses for ocular medication delivery have attracted a lot of attention because they have the potential to improve ocular bioavailability and patient compliance, both with minimal side effects. However, it is essential not to compromise essential features such as water content, optical transparency, and modulus to attain positive in vitro and in vivo outcomes with respect to a sustained drug delivery profile from impregnated contact lenses. Aside from difficulties like drug stability and burst release, the changing of lens physico-chemical features caused by therapeutic or non-therapeutic components can limit the commercialization potential of pharmaceutical-loaded lenses. Research has progressed towards bioinspired techniques and smart materials, to improve the efficacy of drug-eluting contact lenses. The bioinspired method uses polymeric materials, and a specialized molecule-recognition technique called molecular imprinting or a stimuli–responsive system to improve biocompatibility and support the drug delivery efficacy of drug-eluting contact lenses. This review encompasses strategies of material design, lens manufacturing and drug impregnation under the current auspices of ophthalmic therapies and projects an outlook onto future opportunities in the field of eye condition management by means of an active principle-eluting contact lens.

## 1. Introduction

The concept of delivering pharmaceuticals through contact lenses was introduced by Otto Wichterle and Drahoslav Lim in their influential Patent 3,220,960, which was granted in 1965 [1]. This breakthrough paved the way for the advancement of contact lenses made from poly (2-hydroxyethyl methacrylate) (PHEMA). The inventors described the diffusion of antibacterial boric acid through PHEMA devices [2]. Following this discovery, some early exploratory experiments were centered on the soaking method of immersing the contact lens in a drug solution, then inserting the lens into the eye [3]. Since then, contact lenses with drug release for therapeutic purposes have become one of the most promising platforms for improving bioavailability of ophthalmic drugs, and are now emerging as the next-generation treatment of corneal and ocular surface diseases (OSDs).

Topical, intracameral, subconjunctival, retrobulbar, and systemic routes of administration are some available methods for the treatment of eye conditions. Topical administration, which includes both ointments and more fluid medications like eye drops, has the advantage of non-invasivity and is a frequently employed method of treating anterior segment illnesses, accounting for more than 90% of all ophthalmic formulations [3]. Because of the high turnover rate and restoration time of the tear film, topically applied eye drops are quickly drained into the nasolacrimal duct and removed by scleral blood and the lymphatic system [4]. As a result, the bioavailability barely reaches 1–5% in the target tissue (the amount of lipophilic drugs delivered to the anterior chamber is less than 5% and less than 0.5% for hydrophilic molecules) [5]. In order to address the issue of restricted bioavailability, frequent administration of eye drops is necessary, which might potentially result in suboptimal patient compliance, particularly in cases of chronicity (i.e., glaucoma, dry eye disease, etc.) [6].

Contact lenses, in this setting, became drug-releasing devices that essentially transcend the aforementioned constraints while also providing additional benefits. In recent years, there has been significant research conducted on the utilization of contact lenses as drug-carriers. This is mostly due to their potential to enhance bioavailability, leading to improved efficiency and therapeutic compliance [6]. Depending on the materials of choice, contact lenses may be classified as hard or soft. Hard lenses are often constructed of a rigid gas-permeable material, while soft contact lenses are made of a flexible hydrogel that permits oxygen to cross through to the cornea [7]. They provide pain relief, hydrating the corneal epithelium, improving corneal wound healing, and acting as a buffer between the cornea and the external environment. Bullous keratopathy, corneal erosions, corneal epithelial abnormalities, and post-operatory diseases such as post-keratoplasty and post–laser vision correction are all treated using therapeutic lenses [8].

Designing enhanced therapeutic contact lenses is quite a tricky task, since most of the paths chosen by scientists when researching better materials often includes taking the trial-and-error approach. Trial-and-error means time-consuming experiments which can lead to nothing newly discovered the majority of the times. It also includes costly procedures and repetitive tasks that waste valuable time and also financial resources that could be used in other better ways within the R&D process.

Computer simulations, on the other hand, have proved to be a good fit for choosing and adjusting appropriate biomaterials when trying to develop new medical devices. They are also a match for imaging in bio-nanotechnology, spanning the length and temporal scales up to several nanoseconds for atomistic approaches and much longer for coarser descriptions. Recent advancements in the field of computer science, coupled with the integration of diverse approaches for conducting multi-scale in silico research, have facilitated the advancement in the development of novel medical devices that exhibit enhanced features [9,10,11].

A combined approach, computer simulations and real design, allows scientists to adjust certain parameters based on the most promising in silico results in order to meet real-world clinical needs. Such achievements are being realized primarily because molecular modeling is able to generate nanoscale pictures with atomic and even electronic resolution, to predict the nanoscale mechanisms of new biological and inorganic materials for biomedical applications [12,13].

The fast development of computer simulation and complex actual design/research and technology provided a feasible way to analyze complex system behavior and evaluate the intervention effects, and resulted in great progress in therapeutic contact lenses [14].

The aforementioned combined approaches allowed the obtainment of effective designs able to preserve effective concentrations of drug on the ocular tissue, and are becoming the most common method of ophthalmic drug administration [8].

Accordingly, the newly developed TCLs are successful for corneal and OSD care; however, contraindications and complications such as infective keratitis, corneal anesthesia, corneal hypoxia, corneal allergies and inflammation, unsustained drug release, and poor lens fit are some of the main challenges that still have to be considered [8].

Numerous recent review articles that offer an extensive overview of the production and application of contact lenses (CLs) have been published. These papers provide a detailed examination of the most successful materials, preparation processes, and current advancements in the field of ocular surface disease (OSD) care [15,16,17].

Thanks to rapid advancement in the field, complex TCLs which display drug release over an extended period based on new materials obtained by sophisticated fabrication techniques were lately proposed. This updated review presents the various materials and techniques to fabricate and deliver drugs through CLs and goes one step further, examining their advantages and shortcomings. First, the different CLs materials judiciously picked to control the release profile of relevant drugs are addressed. Further on, manufacturing methods for producing CLs and TCLs that contain and release active ingredients followed by the latest scientific results which demonstrate the potential of these ground-breaking noninvasive drug delivery systems are discussed in depth. Another main goal is to overview the latest advances in the treatment and/or prophylaxis of eye pathologies (i.e., glaucoma, cataract, corneal diseases, and keratoconus) using TCLs. In addition, this review discusses the latest developments in novel therapies for the keratoconus condition. Last but not least, the current paper gives an overview of the future perspectives and challenges in the field of TCLs for OSDs.

## 2. Material Design and Contact Lense Fabrication

### 2.1. Main Materials Used in Contact Lense Fabrication

Current ocular treatments are outperformed by contemporary contact lense materials in terms of drug delivery. It is necessary to further modify the base polymer structure of these devices in order to support and enhance the therapeutic outcome [18]. The most prevalent alteration techniques are molecular grafts, particle encapsulation, and soaking [3,19].

Double-network/interpenetrating hydrogels and pH-responsive polymers are two classes of materials that have experienced rapid growth. Although there has been considerable interest in areas such as liquid crystal contact lenses [20,21,22], the number of research articles in these areas remains relatively low in comparison to other technologies.

Double-network/interpenetrating hydrogels create a new composite gel by linking the gel networks of two gels [23]. Instead of a typical copolymer gel, the network of one gel is interlaced with the network of the other gel, as long as they have the same functional groups [24]. Figure 1A exhibits a schematic representation of different hydrogel structures, demonstrating the main structural difference between traditional single polymer networks, a double-network and a cross-linked double network. Temperature- and pH-responsive hydrogels are based on macromolecules that change conformational shape when certain acidic/basic variations (e.g., caused by inflammation [25]) or temperature stimuli are applied [25,26,27]. Figure 1B,C provides a schematic representation pH-responsive hydrogels in both an acidic and basic medium.

The terms “hard” and “soft” are frequently employed as broad labels for contact lenses. Hard contact lenses are gas-permeable durable devices which are often mislabeled as rigid gas-permeable lenses (RGPs); a clear distinction should be enacted, since numerous counterexamples could be brought forward [30,31,32]. A soft contact lens (SCL), on the other hand, is made of polymer, with higher water content that favors oxygen permeability. Because of their flexibility, SCLs adapt to the curvature of a patient’s eye significantly quicker than their stiff counterparts. SCLs can be disposed of each day, week, or month [33,34].

Nonetheless, such broad definitions of contact lenses can provide some insight into their material qualities, but not always. Materials used in hard and soft lenses frequently overlap, but there are differences in the chemical composition, structuration, gel network, water content, etc., augmenting the spectrum of feasible contact lenses and their attributes [19]. Table 1 summarizes the main polymer materials used for the fabrication and the CL features derived from the material used.

In terms of drug accommodation, silicone hydrogels and other hydrogels are believed to be the optimal materials for the fabrication of contact lenses [35]. Early hydrogel contact lenses made by polymerizing 2-hydroxyethyl methacrylate (HEMA) did not let enough oxygen into the eye. The addition of different hydrophilic monomers to HEMA polymer formulations increased the water content and made it easier for oxygen to pass through. However, HEMA-based hydrogel CLs are only suited to be worn at most on a weekly basis [35]. Silicone hydrogel lenses, on the other hand, allow an important amount of oxygen to pass through and can be worn for 29 days [36].

Contact lenses facilitate drug flow to the eye surface for more than 30 min, which, compared to the eye drop delivery of roughly two minutes, is highly advantageous. This means that the bioavailability of the drug on the cornea is superior, because contact lenses deliver the loaded bioactive drugs for a longer extent than conventional eye drop devices. As a result, the amount of drug in the body’s bloodstream is cut down, and so are the possible side effects. Drug-filled contact lenses can be designed to be worn for longer periods, which reduces the number of times the patients need to be administered with medication [37].

The first hydrogel contact lens with a medication infusion was developed in 1965. They studied homatropine (1% aqueous concentration solution) in which they incubated lenses and observed that the effect of pupil dilation was longer in patients than they would have with just eye drops. After that, there was interest in using contact lenses to administer pilocarpine in glaucoma management, and an increasing number of studies started looking at contact lens effectiveness using in vitro and in vivo tests to treat different ocular disorders [17].

The medicine delivered by contact lenses has a minimum corneal bioavailability of 50% under ideal circumstances. A mathematical framework for the release of bioactive agents through contact lenses has been developed in an effort to accurately predict the corneal bioavailability; however, it ignores some contact lens concerns, such as swelling and the interaction with the lodging network [38]. It has been proven that the rate of radial diffusion, the rate of drug equilibrium between the contact lens and the tear film, and the ratio of corneal drug uptake are the three most influential parameters on corneal bioavailability. Predictions place corneal bioavailability between 50 and 70 percent [35].

Poly methyl methacrylate (PMMA), cellulose acetate butyrate (CAB), and siloxy methacrylate (SMA) paved the way in stiff contact lense fabrication. After Mulle and Ohring’ first rigid PMMA contact lenses in 1936, the Food and Drug Administration (FDA), in 1978, approved CAB as an alternative to PMMA due to its superior gas permeability. SMA sparked the development of a new generation of contact lenses with exceptional stiffness and gas permeability by combining a methacrylate backbone with siloxane groups. Surface wetness was a prevalent concern due to SMA’s high lipophilicity, which resulted in surface scratches and superficial lipid fouling [35].

**Table 1 jpm-13-01571-t001:** Comparison of CL features fabricated based on different materials.

Lens Type	Material (Monomer)	Features	References
Rigid	Methyl methacrylate	-low permeability -stiff	[32]
	Cellulose acetate butyrate	-superior gas permeability to PMMA -stiff	[35]
	Siloxy methacrylate	-exceptional gas permeability -low surface wetness -lipid surface deposits	[33]
	Fluoro-siloxymethacrylate	-gas permeability higher than PMMA -improved wettability -no considerable balance of clinical advantages over PMMA	[39]
Soft	Hydroxyethyl-methacrylate	-not enough O_2_ permeability	[31,34]
	N-vinyl pyrrolidone	-high water content -increase in the relative evaporation rate of water -beneficial effect on drug loading and release	[40]
	HEMA-co-NVP	-high water content when compared to pure polymer -higher O_2_ permeability -improved drug loading and more optimal drug release	[31]
	HEMA-co-HEMA-co-MPTS	-excellent cationic drug loading -improved drug release (Gatifloxacin and Moxifloxacin) compared to commercial etafilcon A and polymacon and eye drops	[41]
	TRIS-DMA-NVP-HEMA	-best balance of oxygen permeability, equilibrium water content, hydrophilicity and reduced protein film formation compared to simpler formulations	[33]
	TRIS-NVP-MAA-PEGMA	-the overall oxygen permeability, friction coefficient, water absorption capacity, contact angle, modulus and protein adsorption are superior to some of the commercial contact lenses (e.g., Acuvue Advances or Cooper vision).	[42]

Soft contact lenses, compared with the hard ones, have the benefit of being more comfortable and biocompatible. Because the cornea mostly receives oxygen dissolved in water, a hydrogel contact lens’ oxygen permeability is crucial, and so it is its water content that governs it. Hydrophilic monomer N-vinyl pyrrolidone (NVP) is commonly used to increase the soft contact lens water content [40]. Yet, the evaporation rate of water from NVP networks is higher, which can result in lens surface roughing, creating discomfort to the wearers; but, on the other hand, rougher surfaces can help with drug loading and improving the release of active molecules [19]. Hydrogels with microstructural changes had more water content, which improved drug loading and resulted in a more optimal release profile. For this reason, in one study, by adding NVP and 20–40% *v*/*v* water to the HEMA contact lens, pores were generated within the hydrogels during the polymerization step [31]. Furthermore, when poly(vinyl pyrrolidone)is copolymerized with HEMA, an increasing surface hydrophilicity and reduced surface friction was found. Conversely, ion ligand copolymerization of the highly negatively charged anionic methacrylic acid (MAA) monomer with 3-Methacryloxypropyltris(trimethylsiloxy)silane (MPTS) and HEMA can improve cationic drug loading, i.e., Gatifloxacin (GFLX) and Moxifloxacin (MFLX) antibiotics [41]. In an antibiotic drug model, the release from the new MAA-MPTS-HEMA lenses was found to be considerably higher throughout 72 h than that of the commercial Etafilcon A^®^ and Polymacon^®^ (*p* < 0.01) lenses. Furthermore, the corneal and aqueous humor bioavailability surmounted that of the eye drop groups. The carboxyl groups of MAA, on the other hand, attract positively charged proteins like lysozyme, resulting in substantial protein fouling.

The corneal edema caused by overnight lens usage, which promotes excessive water build-up (favoring bacterial growth), was addressed by adding silicone to the hydrogel which increased oxygen permeability and decreased the equilibrium water content. Silicone hydrogel contact lenses are often made out of (i) silicone-based polymers such as poly-dimethylsiloxane (PDMS), tris-trimethylsiloxysilyl (TPVC), tris(trimethylsiloxy)- methacryloxy-propylsilane (TRIS), or other siloxane macromers, and (ii) hydrophilic monomers like HEMA, N, N-dimethylacrylamide (DMA), or NVP. TRIS-DMA-NVP-HEMA contact lenses’ optimum balance of oxygen permeability, water content equilibrium, hydrophilicity, and reduced protein film formation overcomes most composition constraints [33]. Thus, by combining materials with desired properties and the right fabrication procedure, soft contact lenses with customized features can be made. Another study examined a TRIS-NVP-MAA-PEGMA complex. The hydrogel presents a nice balance between key features when compared to some commercial contact lenses, reporting a hydrophilicity (contact angle and water absorption capacity) similar to that of ACUVUE Oasys^®^ and ACUVUE Advance^®^, an oxygen permeability spanning 45.3 to 73.0 barrers, which is superior to 1-Day ACUVUE^®^ (21.4 barrers), Biomedics XC^®^ (44 barrers), and Biomedics 38^®^ (8.4 barrers), an elastic modulus strongly dependent on the PEGMA ratio, spanning 1.42 MPa to 0.69 MPa, while CIBA Vision^®^ moduli are higher (1.0–1.52 MPa), a friction coefficient varying between 0.288 and 0.075, as the PEGMA content increased, while that of Clariti^®^ 1 day, (CooperVision^®^, Pleasanton, CA, USA) is 0.069 and shows protein adsorption [42].

Despite proof of the improved performance of these novel materials, additional work is required to bring them to the stage of commercialization. The greatest market hurdle is the cost of clinical trials and the manufacturing needs.

The development of contact lens materials has undergone a constant evolution (Figure 2), progressing to the present day, with significant advancements occurring with increasing frequency, as Chadhari et al. also described [43]. Not long ago, the market for contact lenses was mostly controlled by silicone hydrogel lenses, accounting for 64% of the market share, while hydrogel lenses constituted 22% of the market. The utilization of RGP materials has experienced a gradual decrease over a period of time, accounting for just 1% of the overall market [44]. In this context, it was imperative to address the issue of the latest emerging directions that could impact the market development of next-generation TCLs.

In order to provide a more comprehensive depiction of the environment and offer a glimpse into the potential future applications of contact lenses, Papas examined the patent literature within the timeframe from January 2014 to February 2017. A search was performed targeting all publications containing the phrase “contact lens” in their titles, and the most knowledgeable results were surveyed [45]. In a similar fashion, we investigated the number of patents for “contact lens”, “silicone hydrogel contact lens”, “hydrogel contact lens” and “rigid contact lens” for January 2014–December 2016, February 2017–December 2019, January 2020–December 2022 and January 2023–October 2023, in order to assess the recent trends in prominent research and development conducted in the field. Google Patents [46] was the tool employed, and the results are plotted in Figure 3.

It was interesting to observe a slight increase in the incidence of novel RGPs, considering the market trends. A more in-depth analysis of the data and a brief literature survey highlighted a new emergent inclination towards “smart contact lenses” which are a novel category of ocular devices capable of monitoring, among others, diverse health parameters, including glucose levels [47,48,49], lactic acidosis [50], and intraocular pressure [51,52,53]. In addition, wireless communication with portable electronics [54,55], and virtual reality/augmented reality technology [56] can be integrated, so enabling users to obtain real-time information and supplement their visual perception. At the present time, it seems that the design of such devices better suits the spectrum of rigid CLs [57,58].

Nonetheless, attempts to accommodate the attributes of smart and flexible in one have also been reported [59,60]. Backed up by the growing effort and resources poured into portable electronic devices, smart CLs of both flexible and rigid nature stand now in the spotlight in clinical trials [61,62,63,64,65,66]. As they are more and more explored by transdisciplinary experts, they are prognosticated to be truly multifunctional and, apart from visual acuity monitoring [67] and on-demand drug administration [68], from within the process of development to lower or solve some of the limitations of the material itself that hosts the smart components [58].

### 2.2. Manufacturing Methods

Manufacturers have developed several methods for making contact lenses with active molecules. Lathe-cutting, spin-casting, and cast-molding are the most common contact lens manufacturing methods. Different manufacturing methods must follow material-specific methodologies. These material manufacturing phases may alter lens properties. Making lathed lenses requires solid dehydrated buttons bulk polymerized over a long period of time [69]. Computer-controlled spin casting injects monomers into a spun mold. The mold shape determines the lens front surface. The mold’s spin speed, the surface tension and friction forces between the mold and the polymer, and gravity all affect the lens’ back surface [69]. Spin casting is faster than button manufacturing in lathing, but the lens takes up to 60 min to polymerize. Monomers are inserted between two castings to produce disposable lenses in a relatively efficient procedure that is simple, rapid, and ideal for bulk manufacturing because of its advantages [69].

These methods have limitations in materials, design flexibility, and intricate geometry, and this is why personalized, multi-functional, smart contact lenses may benefit from additive manufacturing (AM) [70]. AM, also known as a 3D printing technique, bears the promise of computer-aided design (CAD) accuracy and undemanding dimensional control of the architectural features of the device, which could be readily customizable. The technology is appealing, since it is time and money saving while also requiring little post-processing of the 3D prints. Compared to other traditional manufacturing technologies, AM allows for precise item replication and the production of multiple products at the same time [71,72]. Selective laser sintering (SLS), fused deposition modeling (FDM), photocuring stereoscopic printing, a stereolithography apparatus (SLA), and digital light printing (DLP) are some of the 3D printing processes used for CL fabrication [73]. Due to the excellent resolution of light-curing-based 3D printing methods (such as SLA and DLP) which enable the deposition of fine printable layers, these techniques are frequently used in the fabrication of such optical devices [74,75].

## 3. Embodying Bioactive Molecules into Contact Lenses

Several strategies have been developed for loading drugs into contact lenses e.g., soaking in a drug solution; a layered structure for drug-eluting or drug-barrier coatings; the incorporation of functional molecules, molecular imprinting, a supercritical fluid method, and colloidal nanoparticles [76]. Table 2 summarizes the main strategies currently used to embody active molecules within contact lenses.

### 3.1. Soaking Method

This is the simplest and least expensive way to load drugs into contact lenses, by soaking [77]. It has worked well for loading ophthalmic drugs like timolol, dexamethasone, pilocarpine, pirfenidone, aminoglycosides, and fluoroquinolones into contact lenses [35].

The process of soaking can be accomplished by two methods: either by introducing medications into the contact lens or by adhering them to the polymer matrix. Drug distribution into contact lenses is facilitated by molecular diffusion, which occurs due to a significant concentration disparity between the soaking solution and the water content of the contact lens. Therefore, it can be inferred that the primary mechanism by which medicines are released from the lens is molecular diffusion. The release of medications with a low molecular weight, ranging from 300 to 500 Da, typically occurs within a span of minutes to hours [78]. Drugs with a high molecular weight, like hyaluronic acid, have a hard time getting into the water channels of a contact lens and staying on the surface [79]. This is because different types of contact lenses have a different affinity for drugs. This means that the type of lens also affects the amount of drugs that can be put in and taken out. Even though soaking is a good way to embed bioactive molecules, drawbacks such as burst release have been highlighted, and many hydrogels release the whole amount of drug in a few hours.

To address this issue, putting a diffusion barrier in the polymer matrix can help the drug stay in the lens for a longer time. However, the choice of a diffusion barrier needs to be addressed carefully, because it needs to be safe and clear, and allow oxygen to pass through the contact lens [78]. Vitamin E is a lipophilic liquid possessing potent antioxidant properties. It is also biocompatible and could have therapeutic benefits. Vitamin E does not impact light transmissibility, since it has a shorter wavelength than visible light. It has been used as a diffusion barrier, making it hard for drug molecules to spread in the soaking method, affecting how ions and oxygen move through the gel. Most vitamin E agglomerates that form are in the hydrophilic domains of the gel. Even though vitamin E-aggregates lengthen the time it takes for a drug to be released, hydrophilic drugs move through a vitamin E-filled hydrogel in a different way than hydrophobic drugs. In a recent study, commercially available ACUVUE TruEye^®^ (New Brunswick, NJ, USA) CLs were loaded with vitamin E barrers for extended and simultaneous release of timolol and dorzolamide, hydrophilic drugs usually prescribed in glaucoma conditions. Extended release of up to 2 days for both timolol and dorzolamide drugs is achieved by loading lenses with vitamin E barrers. Both timolol and dorzolamide revealed superior intraocular pressure (IOP) reduction, compared to eye drops (about 6-fold lower drug loading) [80]. In another recent study, same commercially available CLs were modified with vitamin E by the soaking method, although this time vitamin E was dissolved in ethanol–water solutions to mitigate the extent of swelling. After that, the transport of timolol, a hydrophilic glaucoma drug, was assessed. The integration of the vitamin E barrier into contact lenses was achieved through the immersion of the lenses in aqueous and ethanolic solutions of vitamin E. The aforementioned protocol was successful in extending hydrophilic drug release, and lowered the swelling degree; this is advantageous, as it reduces the risk of lens damage when loading vitamin E [81].

Despite obvious advantages of using the vitamin E barrier to control the drug delivery, limitations related to the reduction in ions and oxygen permeability, the alteration of CL mechanical properties and protein adsorption, due to the hydrophobic nature of vitamin E, have to be considered [79].

### 3.2. Incorporation of Functional Molecules within Contact Lenses

Increasing affinity is advantageous, because it will make it easier for drugs to be loaded and released, so other ways have been suggested. In one example, ion components are added to contact lenses before they are polymerized. Then, the lenses are soaked in a drug solution, which makes them more likely to interact better with drugs. In some studies, copolymerization was used to load all ionic components with acrylic/vinyl groups into the hydrogel matrix (e.g., HEMA, methacrylamide propyl-trimethylammonium chloride (MAPTAC), 2-methacryloxyethyl acid phosphate (MOEP) and MAA) [82]. The hydrogel stated earlier, which possesses a cationic function on its side chain, has demonstrated the capability of retaining the anionic drug azulene through an ion-exchange reaction. Furthermore, the composition of this complex has effectively prevented changes in size and facilitated controlled drug delivery [82]. Dexamethasone 21-disodium phosphate-filled HEMA contact lenses with the cationic surfactant, cetalkonium chloride, were designed. Dexamethasone 21-disodium phosphate, an ionic drug, can then be stuck to the charged surfactant that covers the surfaces of the polymer matrix. Surfactants can be added to either pre-monomer mixtures or pre-made lenses. When 10 percent of the lens is made up of surfactant, the drug stays in the lens for 25 times longer than in the ACUVE TruEye^®^ contact lens. Good wettability, excellent transparency, and low protein adsorption are other features that characterize the HEMA/cetalkonium chloride lenses [83].

**Table 2 jpm-13-01571-t002:** The main strategies currently used to embody active molecules within contact lenses.

Lens Type	Commercial Name/Monomer Type	Features	Drug	Drug Loading Techniques	Ref.
1	ACUVUE TruEye^®^ CLs with vitamin E barrers	-vitamin E modification increases the release duration of both drugs to about 2 days -lesser dosage of medication in comparison to the use of eye drops has been observed to result in a reduction in intraocular pressure	timolol and dorzolamide simultaneously loaded	Soaking Method	[80]
2	ACUVUE TruEye^®^ CLs with vitamin E barrers	-effective at sustaining release of timolol -reduced swelling, which reduces lens damage	timolol	Soaking Method	[81]
3	HEMA/MAPTAC/MOEP/MAA	-prevents the size change -efficient drug delivery	azulene	Incorporation of Functional Molecules	[82]
4	HEMA/ cetalkonium chloride	-drug release duration (50 h) -good wettability -low protein absorption -excellent transparency of lenses	dexamethasone 21-disodium phosphate	Incorporation of Functional Molecules	[83]
5	HEMA/DMA/TRIS/CDs	-improves the water solubility of natamycin -more effective in delivering natamycin	natamycin/methacrylated beta-cyclodextrin (Mβ-CD) natamycin/methacrylated 2-hydroxypropyl-β-cyclodextrin	Incorporation of Functional Molecules	[84]
6	HEMA/HEMA–co-GMA/α-, β- and γ-cyclodextrins functionalized	-reduced protein sorption -HEMA γ-cyclodextrins has the highest loading for miconazole -sustained miconazole delivery for over 14 days -high efficiency against biofilm formation	miconazole	Incorporation of Functional Molecules	[85]
7	poly-CDs-HEMA	-high drug doses loaded -sustained drug release for 6 days	ethoxzolamide	Incorporation of Functional Molecules	[86]
8	MAA and methacrylamide (MAm) functional comonomers	-atropine release for up to 72 h -good balance of light transmission, water content, and contact angle	atropine	Molecularly Imprinted	[87]
9	MAA/HEMA/EGDMA	-acyclovir-imprinted hydrogels were not effective in terms of drug loading -valacyclovir-imprinted hydrogels has a sustained release profile for 10 h -relevant amount of valacyclovir is accumulated in the cornea -promising for delivery to the posterior segment	acyclovir valacyclovir	Molecularly Imprinted	[18]
10	Hilafilcon B commercial	-higher flurbiprofen loaded -sustained release profiles	flurbiprofen	Supercritical fluid (SCF)-assisted molecular imprinting	[88]
11	HEMA/MAA EGDMA/Prednisolone loaded PLGA nanoparticles	-slow drug release of drug over 24 h -release of 10.8% encapsulated drug -insignificant changes in light transmission, wettability, and hydration by loading Prednisolone-loaded PLGA nanoparticles	Prednisolone	Colloidal nanoparticles	[89]
12	Dailies AquaComfort PLUS^®^	-inhibit/eradicate the formation of Pseudomonas aeruginosa and Staphylococcus biofilm	Ozodrop^®^ Ozodrop^®^ gel	Liposome	[90]
13	DMA/siloxane/NVP/EGDMA/HEMA loaded Pluronic^®^ F-68/gatifloxacin	-Pluronic^®^ F-68 improves the drug uptake and sustained drug delivery -excellent optical transmittance, swelling and mechanical features	Gatifloxacin/Pluronic^®^ F-68	Micelles	[91]

Cyclodextrin-based (CD) contact lenses have been widely used in the pharmaceutical field to deliver hydrophobic drugs. CD lenses are praised for their high potential to increase drug bioavailability and stability. Contact lenses are made from different monomers that have reactive double bonds when they are mixed together with free radicals. By copolymerization, CDs made from acrylic or vinyl can be turned into drug–CD complexes that can be put into contact lenses. CDs loaded with a natamycin effect were assayed in silicone hydrogel contact lenses fabricated from DMA, TRIS, and HEMA contact lenses. The study revealed that both silicone hydrogel and HEMA contact lenses had an enhanced drug release capacity, with a notable emphasis on the TRIS formulation [84]. Also, adding CDs to preformed polymer networks HEMA and HEMA-glycidyl methacrylate (GMA) (the reaction of polymer glycidyl groups and the hydroxyl groups of CDs) strongly impacts the phenomenon of protein deposition (a high decrease) and the affinity of the network for loading miconazole (a high increase). Furthermore, constant delivery was observed for over 14 days and completely prevented Candida albicans biofilm formation, as indicated by the in vitro microbiological test [85]. In addition, direct crosslinking of CDs with the contact lens materials has been looked into. Poly-CDs that were made by crosslinking with citric acid were used as carriers for ethoxzolamide in one study. Then, they were loaded into a HEMA-based matrix. The poly-CDs-HEMA composition was able to load higher doses of the drug and sustain its release for 6 days [86].

### 3.3. Molecular Imprinted Contact Lenses

The molecular imprinting technique ranks as one of the most advanced technologies for embedding drug templates into contact lenses. During the process of polymerization, voids are created inside the structure of the contact lens, which can effectively accommodate medication molecules. The soaking process is a viable approach for loading drugs into the lens, due to the lens’ cavities exhibiting a strong affinity for drug molecules. The absence of functional monomers will result in the absence of imprinting. However, if there are too many functional monomers the drugs stay in the material instead of being released. Conversely, cross-linking agents can make contact lenses more durable and less likely to swell in water, but too many of them can make the network structure stiff, and slow down the rate at which drugs can leave the lenses [92]. In a recent study by Zhao et al., molecularly imprinted hydrogel CLs were prepared for atropine delivery, with non-imprinted hydrogel (for control) and MAA and methacrylamide (MAm) functional comonomers. MAA/MAm exhibited an atropine release time of up to 72 h, combined with good surface wettability, biocompatibility, light transmission, and water content, which recommends them as promising systems for efficient ocular drug delivery for myopia [87]. Another study was conducted to fabricate hydrogels, both imprinted and non-imprinted, containing different proportions of the methacrylic acid monomer, HEMA, and EGDMA, with the purpose of facilitating the administration of acyclovir (ACV) and valacyclovir (VACV) medicines. Acyclovir (ACV) and valacyclovir (VACV) are considered the primary treatment options for ocular keratitis caused by the herpes simplex virus. MAA/HEMA/EGDMA-ACV-imprinted hydrogels were not efficient for drug loading and release. Conversely, a high affinity of the hydrogel for VACV was noticed. It is highly probable that this phenomenon is attributed to the electrostatic interactions between the acrylic acid group and the VACV lateral chain. The hydrogels imprinted with VACV exhibit sustained drug release over a period of 10 h, resulting in a significant accumulation of the medication in the cornea. Furthermore, the VACV-imprinted hydrogels display swelling, light transmission, and mechanical properties similar to those of the commercial contact lenses [18].

### 3.4. Supercritical Fluid Method

The supercritical fluid method (SFM) implies first the drug being dissolved in the supercritical solvent, then being put together with the framework of the contact lens. The utilization of appropriate supercritical fluid solvents presents a practical approach for incorporating both hydrophilic and hydrophobic medicines into contact lenses. Despite the fact that the release profile from lenses loaded through this method is more extended when compared to the one achieved by the widespread soaking method, the burst release is a common disadvantage that has not been overcome. However, the coupled method of supercritical fluid-aided molecular imprinting is believed to substantiate two interests: the larger amounts of drugs that can be loaded into soft contact lenses and the targeting of an extended time of release [88]. An example of the use of the supercritical fluid method is to load flurbiprofen into Hilafilcon B^®^ contact lenses using supercritical fluid CO_2_. The release profile of the drug from the contact lens loaded via SFM versus the one loaded conventionally confirmed the hypothesis, i.e., larger amounts of drugs loaded and an unsustained release of the drug. Additional investigation has indicated that the utilization of the supercritical fluid method yields less favorable outcomes. The most notable achievement in this regard has been the extension of the drug retention period to several hours [17,93]. These observations should be carefully considered, as more advanced inquiry is needed. To surmount such limitations, a combined approach of supercritical fluid and imprinting was proposed. Yanez et al. anticipated that, by combining the aforementioned methods, limitations related to loading the drug in preformed CLs and the molecular imprinting in which the drug has to be selected before polymerization might result in drug-tailored networks. Commercial Hilafilcon B^®^ CLs were treated using SFM for flurbiprofen loading. The treated SFM CLs were able to load flurbiprofen to a higher extent than when using water-based methods. The interaction of flurbiprofen–Hilafilcon B^®^ CLs directs the formation of specific cavities able to chemically and structurally recognize flurbiprofen, and is believed to be responsible for higher flurbiprofen sorption and for the more sustained release profile [88].

### 3.5. Colloidal Nanoparticles

Colloidal nanoparticles have been used a lot in ocular delivery, since they can support extended drug stays on the cornea. Drugs can also be put inside colloidal nanoparticles to keep them from being broken down by enzymes in the ocular environment. Nanoparticles loaded into a contact lens matrix make a drug delivery system that lasts longer than either the nanoparticles or the lens alone [4]. Within the delivery system, the drug undergoes transportation via nanoparticles in order to access the contact lens matrix. Subsequently, it traverses the hydrogel matrix to reach the ocular tissues. Various forms of nanoparticles, including polymeric nanoparticles, liposomes, micelles, and emulsions, have been effectively incorporated into contact lenses. In addition to the aforementioned drug-laden nanoparticles, it is also possible to incorporate non-medicated nanoparticles into contact lenses. Silver nanoparticles possess potent antibacterial properties due to their ability to deactivate enzymes and impede the process of DNA replication [35]. ElShaer and coworkers investigated the ocular administration of Prednisolone loaded in poly (lactic-co-glycolic acid) (PLGA) nanoparticles, prepared using the single emulsion–solvent evaporation method. Prednisolone-PLGA nanoparticles presented an average particle size of 347.1 ± 11.9 nm, with a polydispersity index of 0.081. CLs fabricated using HEMA (80%), MAA (19%), and EGDMA (1%) were loaded onto Prednisolone-PLGA nanoparticles. Nanoparticle-loaded contact lenses exhibited almost unchanged hydration, light transmission and wettability, and showed a sustained drug release over 24 h, while 10% of encapsulated prednisolone was released [89].

Liposomal delivery is another advanced drug-delivery system that increases the bioavailability and encapsulation, and ensures targeted and sustained release. In a recent study, Zerillo and co-workers examine the effect of Ozodrop^®^ and Ozodrop^®^ gel (ozonated oil in liposomes and hypromellose) on the suppression and/or formation of biofilms on DailiesAquaComfort PLUS commercial CLs. The study outcomes indicated that Ozodrop^®^ and Ozodrop^®^ gel have an exceptional inhibitory effect on the eradication and formation of biofilm produced by Pseudomonas aeruginosa and Staphylococcus aureus [90].

Loading drugs within CLs might affect key features such as the optical features, swelling, or contact angle, due to drug precipitation in the polymer matrix. Designing micelles in the contact lens aimed at dissolving drug precipitates seems to be a promising method to overcome the aforementioned challenge. Maulvi et al. fabricated DMA/siloxane/NVP/EGDMA/HEMA CLs loaded with Pluronic^®^ F-68/Gatifloxacin. Pluronic^®^ F-68 was meant to build micelles and provide better uptake and sustained drug release. The gatifloxacin–pluronic-loaded CLs exhibited excellent optical transmittance, swelling, and drug loading capacity and sustained drug release [91].

## 4. Non-Interventional Future Perspectives in Keratoconus

The versatility of the polymeric supports used in the fabrication of contact lenses offers the possibility of incorporating numerous active compounds, carefully paired with specific impregnation techniques within the polymer supports. So far, within this review, we have cataloged a series of contact lenses envisaged as drug vehicle solutions proposed for ocular conditions such as glaucoma, conjunctivitis, cataracts, diabetic macular edema, retinoblastoma, fungal or bacterial inflammations, etc. Nonetheless, an important ophthalmic disorder, keratoconus, with a prevalence estimated at 1.38 per 1000 individuals worldwide [94], has not, to the best of the authors’ knowledge, been addressed from this standpoint by any review paper. However, based on the currently available options for keratoconus treatment, we hypothesize a possible alternative treatment in line with the topic of the present paper.

Keratoconus is a non-inflammatory, degenerative corneal condition that manifests in the thinning and bulging of the cornea. Typical factors that contribute to the biomechanical degradation of the cornea in this disease include alterations in the structure and organization of corneal collagen, anomalies in the extracellular matrix, and apoptosis of keratocytes in the anterior stroma and Bowman lamina [95]. Children and young people are more likely to have the illness, which may worsen with time.

A range of pharmaceutical and surgical interventions have been employed for the treatment of keratoconus. Rigid gas-permeable contact lenses are commonly prescribed as the initial treatment option for individuals diagnosed with keratoconus [96]. Spectacles may help improve eyesight in the early stages of the condition, but as the disease advances, rigid gas-permeable contact lenses are frequently the best option. If the disease worsens, corneal transplantation may be the only option. Multiple contact lens designs and fittings have been developed to address the distinctive demands of this progressive condition. Corneal scarring, extreme thinning, and sensitivity to contact lenses are all recognized as indications for corneal transplantation, also known as keratoplasty [97]. Keratoconus represents the primary cause of keratoplasty among people under the age of 30 residing in high-income nations [98].

In recent years, the therapy options stated above were enriched by the implementation of collagen cross-linking (CXL) with the utilization of UVA light and riboflavin, also known as vitamin B2. This is a novel strategy that tackles stromal instability directly. It is the only treatment for keratoconus that can prevent the disease from progressing, unlike all other treatments [99]. CXL using UVA light and riboflavin is a relatively novel therapy that has been shown to delay disease development in its early stages. When paired with intracorneal ring segments, the improvement in vision is larger than when the segments are used alone [98].

A promising solution to this issue could be the employment of gas-permeable contact lenses paired with riboflavin as the therapeutic agent for controlled and prolonged release on the eye’s surface. Moreover, wearing these contact lenses, and with exposure to sunlight (which includes UVA light), the process of crosslinking could be achieved in time. Figure 4 presents the distinctiveness between the two approaches currently used for keratoconus eye care; rigid contact lens wearing (the traditional standard procedure) and cornea collagen photochemical cross-linking with riboflavin with UVA light. In the CXL approach, corneal tissues have to be removed as riboflavin cannot pass the corneal epithelium (due to high molecular weight).

The corneal stroma is mostly composed of collagen type I, while collagen types III, V, and VI are also present. The corneal stroma has the mechanical strength to build the eye’s anterior coat while preserving the high degree of transparency essential for light transmission. Collagen fibers and fibrils are formed starting from collagen’s precursor, procollagen. Procollagen has two additional peptides, one at each end, which are removed through enzymatic pathways and undergo post-translational modifications to form the collagen molecules. The enzymatic activity of lysyl oxidase is accountable for the oxidative conversion of lysine and hydroxylysine amino acids into their corresponding aldehydes. These aldehydes subsequently engage in condensation reactions with other aldehydes, resulting in the formation of intramolecular and intermolecular cross-links [99].

There are other methods available for inducing further cross-linking, including those such as nonenzymatic glycation, UV light irradiation (mediated or not by photosensitizers), and aldehyde reactions. The primary objective of therapeutic cross-linking was to enhance the rigidity of the cornea and perhaps impede the progression of keratoconus. This was achieved by actively augmenting the extent of the covalent bonding among various constituents of the extracellular matrix, such as collagen type I and proteoglycans, both intermolecularly and intramolecularly [100]. Riboflavin serves as a photomediator, significantly enhancing UVA light absorption when exposed to the corneal stoma. It has been shown that UVA radiation is absorbed by around 30% inside the lamellae of the corneal stroma, but when combined with the photomediator characteristics of riboflavin, this absorption jumps to 95% [101].

After being exposed to radiation, riboflavin undergoes excitation, leading to the formation of a triplet state, which subsequently generates reactive oxygen species. Singlet oxygen and superoxide anions undergo further reactions with nearby accessible groups [100]. It has to be specified that the precise mechanism of the riboflavin/UVA crosslinking reaction is not exactly and fully understood, but there have been multiple suppositions that could be verified. It is plausible that this phenomenon happens through the creation of additional chemical bonds between histidine, hydroxyproline, hydroxylysine, tyrosine, and threonine amino-acid residues [99,101]. Importantly, riboflavin may promote cross-linking of other macromolecules, such as proteoglycans, inside the corneal stroma, either to one another or to collagen molecules [102]. Corneal treatment based on cross-linking collagen using UVA light and riboflavin remains the foundation preventive approach; nevertheless, a significant obstacle in the present research revolves around the augmentation of oxygen availability and the exploration of novel methodologies for enhancing riboflavin permeability throughout the process. Apart from fast oxygen depletion, predominantly when higher amounts of UVA light are used, and inadequate riboflavin penetration through the tissue, challenges related to the depth-dependent riboflavin gradient within the tissue’s [103] extended treatment period, and potential endothelial toxicity in thin corneas have to be overcome. Recently, protocols to simultaneously combine keratoconus progression with correcting the refractive error have greatly evolved, exhibiting great promise in helping clinicians better manage patients [104]. Conversely, recent developments in regenerative medicine and tissue engineering have led to groundbreaking advancements in cellular-level treatments, hence decreasing the need for invasive surgical procedures.

## 5. Conclusions

Contact lenses are increasingly being seen as a tool for specialized ophthalmic therapies that require adequate drug delivery to the eye. As a result, practitioners rely on the development of new lenses to better deliver these therapies, which is dependent on material scientists tailoring new materials. With a significant change in lifestyle, contact lens research for therapeutic purposes has increased dramatically. Developing contact lens platforms for ocular drug delivery is a novel and effective therapeutic approach for a variety of ocular diseases, while avoiding the limitations of traditional eye drops. However, additional research is needed prior to the potential commercialization of drug-loaded contact lenses in order to ascertain their effectiveness, safety, and overall comfort for individuals. Most strategies for introducing drugs into contact lenses, such as drug-soaked contact lenses, vitamin E-modified contact lenses, molecular imprinting, and the supercritical fluid method (or hybrid processes), primarily focus on loading the drug into the contact lens during the manufacturing process. The studies encompassed in this review demonstrate that the primary challenge is in facilitating the prolonged release of ocular medicines from contact lenses. Undoubtedly, soaking represents a cost-effective and often-employed method for integrating active compounds into contact lenses, leading to limited drug absorption and swift burst release. The utilization of contact lenses as a vehicle for pharmaceutical delivery, together with the incorporation of vitamin E as a diffusion barrier and supplementary active component, has demonstrated notable efficacy in prolonging the release of drugs for the treatment of ocular diseases. Furthermore, recent findings have indicated that the integration of drug-loaded liposomes and micelles into contact lenses is a viable approach for mitigating the issue of drug leakage commonly observed in soaking lenses. However, it is important to note that the current body of literature on this topic is limited, necessitating further investigation to substantiate these findings. Although the recently proposed molecular imprinting utilizing supercritical CO_2_ has the potential to overcome some of the drawbacks of conventional molecular imprinting, more research into its application is required. It is critical to emphasize that, apart from facilitating substantial drug loadings and sustained release, an effective strategy for manufacturing therapeutic contact lenses also has to ensure fundamental attributes such the lens’ transparency, water content, and gas-permeability characteristics. In order to prevent the untimely and undesired discharge of the medicated substance, it is necessary to also consider additional stages such as the sterilization, packaging, and storing of therapeutic contact lenses. The rigid and soft CLs have found application also in the treatment of the keratoconus corneal condition, which is a gradual pathology characterized by the thinning and protruding of the cornea, accompanied by a reduction in its rigidity. Although many methodologies are employed in the therapy for keratoconus, non-surgical techniques often endeavor to halt the advancement of the condition and enhance visual acuity. Various therapeutic techniques are employed, including the utilization of hard contact lenses or corneal collagen cross-linking. Riboflavin photochemical corneal collagen crosslinking is an approved procedure by the FDA that aims to enhance the strength of corneal tissue. It is widely regarded as the current benchmark in this field. The conventional corneal cross-linking treatment is presently the most widely utilized and reliable method in clinical settings. In conclusion, new contact lens materials will continue to extend the boundaries of materials science in order to better customize to the needs of an expanding contact lens-using population. Taking into account the fast-paced evolution in personalized medicine and telemedicine, and also the wide availability of outstanding commercial materials, smart contact lenses are deemed to become the next pivot for ophthalmic devices of unprecedented complexity.

## Figures and Tables

**Figure 1 jpm-13-01571-f001:**
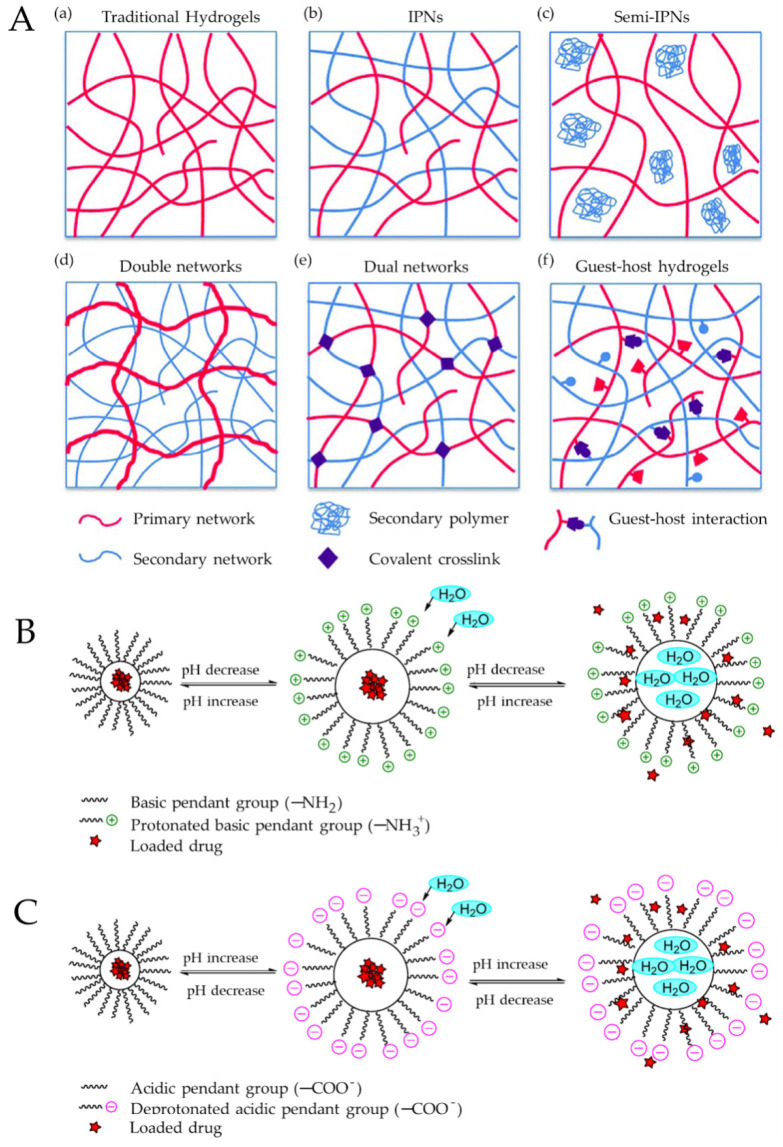
(**A**) Schematic representation of different hydrogels, (a) traditional single polymer networks (b–d) double networks and (e,f) cross-linked double networks. This figure is from an open access journal under a CC-BY license source [28]. (**B**) pH-responsive polymer consisting of amino group (–NH_2_) and loaded drug. In a basic pH environment, the amino groups are ionized and they keep the drug within the polymer network; for acidic pH, the amino groups were protonated (NH_3_^+^) and electrostatic repulsion between the polymer and drug positive groups takes place, releasing the encapsulated drug. (**C**) pH-responsive polymer containing a carboxylic group (–COOH) loaded with drug. In the acidic pH environment, the acidic group is unionized and encapsulates the drug in the polymer matrix. For basic pH, the –COOH group ionizes, increases the electron charge density, and releases the incorporated drug. These figures have been taken from an open access journal under a CC-BY license source [29].

**Figure 2 jpm-13-01571-f002:**
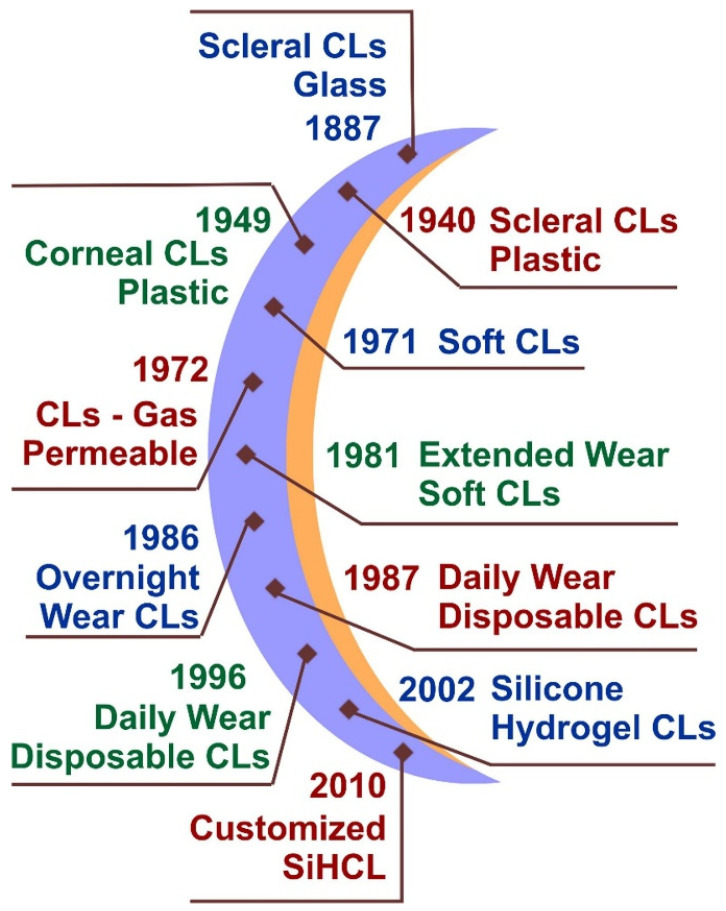
Evolution of contact lenses from the first introduction as scleral CLs made of glass to the introduction of customized silicone hydrogel contact lenses (SiHCLs). Reprinted with permission from Elsevier [43].

**Figure 3 jpm-13-01571-f003:**
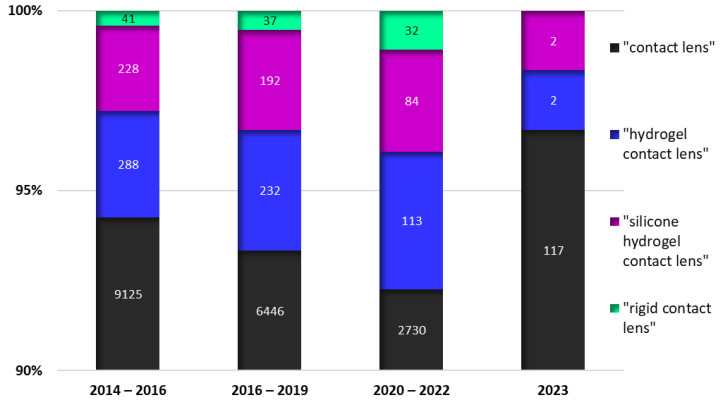
Stacked column representation of Google Patents indexed documents on the topic of CLs.

**Figure 4 jpm-13-01571-f004:**
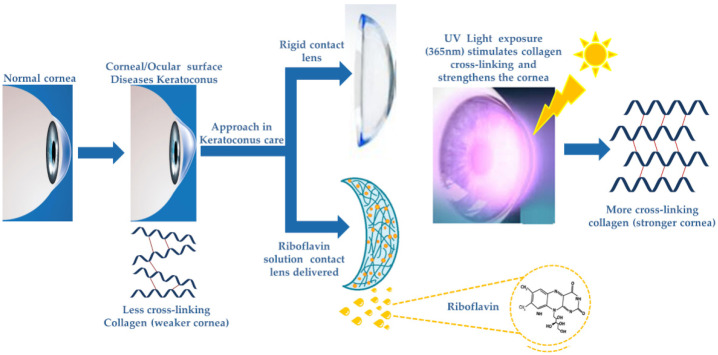
Schematic representation of keratoconus eye; current treatment approaches for keratoconus, i.e., rigid contact lens wear and cornea collagen photochemical cross-linking with riboflavin with UVA light.

## Data Availability

Not applicable.

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
