# Peer review of "Advances in Therapeutic Contact Lenses for the Management of Different Ocular Conditions"

_jpm, 2023, doi:10.3390/jpm13111571_

Round 1

Reviewer 1 Report

Comments and Suggestions for Authors

Title: Advances in therapeutic contact lenses for different ocular condition management

This subject matter holds significant importance and warrants careful examination in the realm of therapy. The reviewer concured with the assertion that this essay presents a significant argument. Nevertheless, the composition exhibits a noticeable degree of repetition, hence necessitating refinement. Additionally, the classification employed within the piece lacks clarity, thereby warranting further improvement. Table 2 looks difficult to read.

Major comments

1.     The quality of figures (Fig 1and Fig 2) in a review article is suboptimal. Kindly enhance the level of quality.

2.     The composition exhibits a noticeable degree of repetition, hence, necessitating refinement and tables.

3.     Backbone definition is complicated and “ACUVUE” look commercial name which is not used (https://doi.org/10.1016/S0142-9612(03)00622-7)

4.     We require data regarding the practical implementation and utilization of the subject matter in question. There is a need to decrease the inventory of contact lenses that are currently not in use.

Minor comments

English correction is urgently necessary and lowering similarity.

1.     Line 37 The inventors described the diffusion of antibacterial boric acid through PHEMA devices, including lenses, for later release to the [1]

2.     Lines 98-114. It is difficult to undersand.

3.     Line 507, CXL? collagen cross-linking (CLX)?

Comments on the Quality of English Language

English correction is urgently necessary.  

Reviewer 2 Report

Comments and Suggestions for Authors

The review article titled “Advances in therapeutic contact lenses for different ocular condition management” discussed the strategies of material design, lens manufacturing, and drug impregnation under the current auspices in ophthalmic therapies and projects an outlook onto future opportunities in the field of eye conditions management by means of active principles-eluting contact lens. The review article is well-drafted, and the content is also good. The following points may be considered before accepting the manuscript for publication:

1.      Line 34: The author needs to revise the sentence, as it is the first sentence of the manuscript, which may not be suitable to start with ‘in their’.

2.      Line 38: Better to provide the reference for the original patent along with a patent number.

3.      Fig. 1 is not readable; the author can replace it with better quality.

4.   If possible, the author can provide a table related to the patented products and clinical trials, which will be more interesting for the readers.  

5.      Fig. 2 is not readable; the author can replace it with better quality.

6.  Conclusion: The author can provide the future prospects of therapeutic contact lenses.

Comments on the Quality of English Language

The author needs to check the language in some places for the grammar. 

Round 2

Reviewer 1 Report

Comments and Suggestions for Authors

After careful revision, major issue has been resolved.